# Evidence of Tribological Adaptation Controlled by Tribosynthesis of FeWO_4_ on an WC-Reinforced Electron Beam M2 Steel Coating Rubbed against a HSS Disk in a Range of Sliding Speeds

**DOI:** 10.3390/ma16031013

**Published:** 2023-01-22

**Authors:** Nickolai Savchenko, Evgeny Fedin, Irina Sevostyanova, Evgeny Moskvichev, Andrey Vorontsov, Sergei Tarasov

**Affiliations:** 1Institute of Strength Physics and Materials Science, Siberian Branch of Russian Academy of Sciences, 634055 Tomsk, Russia; 2School of Non-Destructive Testing, Division for Electronic Engineering, National Research Tomsk Polytechnic University, 634050 Tomsk, Russia

**Keywords:** coating, wear, friction, iron tungstate, mechanically mixed layers

## Abstract

In the present work, the tribological experiments on sliding the electron beam composite M2+WC coating have been carried out with characterization of the sample microstructures and phases both before and after the testing using metallography, SEM, EDS, and XRD. The sliding in the speed range 0.8–3.6 m/s resulted in simultaneous reduction in both wear rate and coefficient of friction with the sliding speed. Investigations showed that such a tribological adaptation was due to the tribochemical generation of lubricative FeWO_4_ and Fe_2_WO_6_ mixed oxides and the generation of a mechanically mixed composite layer on the worn surfaces that consisted of carbide fragments, an oxidized metal matrix, and was lubricated by in-situ formed mixed iron-tungsten oxides.

## 1. Introduction

The study of tribological adaptation mechanisms in new composite materials is necessary for their successful application in various industries [1,2,3,4,5,6,7]. There is a well-known, but insufficiently studied, tribological adaptation mechanism for materials containing elements such as W and Fe that is related to the formation of a tribochemically synthesized FeWO_4_ mixed oxide on a worn surface, which can serve as a solid lubricant due to the presence of crystallographic easy slip planes [8,9,10,11,12]. The tribosynthesis of iron tungstates was found during sliding on a WC-(Fe-Mn-C) composites [10] and then was used for developing a wear-resistant hybrid WC/Y–TZP–Al_2_O_3_ composite with dispersed Hadfield steel particles [9]. Another example of tribological adaptation is tribosynthesis of a copper tungstate CuWO_4_ on the worn surface of W-31 wt.% Cu magnetic pulse consolidated samples at 400–450 °C that also allowed reducing friction [13].

Let us note that in-situ tribosynthesis of mixed oxides other than tungstates may also provide friction and/or wear reduction, i.e., tribological adaptation; for instance, samples consolidated from an immiscible Fe/Cu electro-explosive powder mixture provided friction and wear reduction owing to in-situ tribologically synthesized at 400 °C CuFe_2_O_4_ cuprospinel [14].

It is worthwhile to note that many tungstates are interesting materials not only from the viewpoint of tribology; for example, PbWO_4_ [15], ZnWO_4_ [16], and NiWO_4_ [17] are widely used in scintillators and radiation detectors. Multiferroic behavior of MnWO_4_, i.e., a material that combines magnetism and ferroelectricity, was reported [18]. FeWO_4_ is a promising material for electrochemical capacitors working in neutral water-base electrolytes [19,20]. Iron tungstate Fe_2_WO_6_ was used for making anodes for supercapacitors [21], photocatalysis [22], and as a photoelectrode material [23].

The use of tungsten carbide for depositing hard coatings in combination with metallic powder mixtures can be performed using such methods as electron beam cladding [24,25,26,27,28], laser cladding [29,30,31,32,33,34,35,36,37,38,39,40,41,42,43,44,45,46], plasma cladding [47,48], and hybrid cladding [49,50,51]. The latter is a combination of the above-mentioned methods. The iron-matrix WC-reinforced coating deposited using electron beam cladding [8,24,25,52], laser cladding [38,41,42,43,44,45], plasma cladding [47], and hybrid cladding [51] proved to be very effective for enhancing the wear resistance of the steel components not least due to adaptive structural and phase transformations occurring in the coating in sliding. For instance, the subsurface structural and phase transformations were studied in sliding the WC/30 wt.% Hadfield steel electron beam pulse irradiated coating against a tool steel counterbody [52]. The irradiation and surface melting were carried out using a high-current electron beam with energy density 5–40 J/cm^2^ at accelerating voltages within the 10–40 kV range. The result of such an irradiation was refining both carbide phase and the matrix grains as well as precipitation of metastable M_23_C_6_ and M_12_C carbides. These microstructural changes provided hardening of the coating by a factor of 1.5, friction reduction by a factor of 2 and improvement of wear resistance in comparison with those of as-deposited coating.

The literature source results show that sliding friction experiments are commonly carried out with the use of sliding speeds < 1 m/s and wear mechanisms of the iron-based WC-reinforced composite coatings are described mainly in terms of abrasive and adhesive interaction [25,35,38,39,40,44,45,46]. Tribochemical adaptation aspects of wear and sliding in these systems are commonly ignored, and there are few of them devoted to this phenomenon.

It was demonstrated by the sliding of electron-beam cladding M2 HSS coatings against AISIS 52, 100, or AISIO7 counterbodies that their wear is determined by the structural-phase state of the coatings [8]. The coating rubbed against the AISI O7 counterbody, demonstrating its wear rate increased with the normal load while its coefficient of friction decreased. This type of behavior was interpreted then as tribological adaptation due to in-situ generated lubricative compound, namely iron tungstate that served also as a binder to consolidate the wear debris into viscous mechanically mixed layers (MML). The higher the iron tungstate content, the more stable was the wear rate dependence on time. These examples are evidence that tribosynthesis is a process that combines severe surface deformation with elevated temperatures and therefore allows obtaining compounds even from immiscible metals.

An elemental composition of the MMCs is an important issue in view of tribological adaptation when in-situ forming mixed oxides may not simply mitigate the adhesive interaction between the surfaces but considerably improve the lubrication especially at elevated temperatures. Tungsten is an element that is capable of forming not only hard, wear-resistant carbides but also the FeWO_4_ and Fe_2_WO_6_ oxides characterized by reduced friction, especially at high temperature sliding [9]. However, there is still lack of reliable data of formation of these oxides in sliding the W/Fe containing materials.

In this work, an attempt is undertaken to disclose the effect of sliding speed on the subsurface structural evolution and tribo-oxidizing in both the steel counterbody and the M2 steel WC-reinforced coating.

## 2. Materials and Methods

The powder used for the deposition was composed of: (1) a water dispersion obtained from HSS M2 powder containing (wt.%): C–1%, Cr–4%, W–6.5%, Mo–5%, V–1.5%, Si < 0.5%, Mn < 0.55%, Ni < 0.4%, S < 0.03%, O_2_ < 0.03%, Fe—balance; and (2) 20 wt.% of WC powder. The substrates were 20 × 30 × 200 mm^3^ plates machined from a 0.3 wt.% C steel. The residual air pressure in a chamber during the deposition was less than 0.013 Pa.

The final deposited tracks had dimensions as follows: ~20 mm width and ~3 mm height. The electron beam decay dependence on the pass number has been used in order to compensate for the effect of preheating from the previous pass, avoid overheating, and keep the melting pool area constant at ~100 mm^2^. Corresponding to each of the four passes, the beam power values were as follows: (1) 4050–4300 W, (2) 2900–300 W, (3) 2160–2300 W, and (4) 2000–2100 W. The electron beam characteristics, including spot diameter, scan length, and substrate feed rate, were 1 mm, 20 mm, and 2.8 mm/s, respectively. Each pass deposition time was 80 s accompanied by the same duration interpass period intended for cooling. The resulting coated samples are referred to herein as “M2/W coating”.

The wear tests have been carried out using an automated sliding testing setup and testing scheme consisting of two “blocks-on-disk” (Figure 1). The test chamber was filled with water up to the disk’s bottom part to provide cooling. Therefore, some amount of water was pulled into the contact zone during the disk rotation.

The sliding speed values were 0.8 m/s, 1.2 m/s, 2.4 m/s, and 3.6 m/s with a normal load of 100 N.

The counterbody ∅50 mm and 12 mm of thickness disk was machined from AISI 52,100 HRC 63…65 steel (see Table 1).

Each test run was performed after finishing the running-in stage, and then four experiments were carried out with a sliding path length of 2 km, irrespective of the disk rotation rate. The wear rate was determined by relating the worn-out volume to the sliding path length. The subsurface microstructures were examined using the cross section metallographic EDM cut, ground, and polished views.

The subsurface structure and phases of the M2/W coating and AISI 52,100 counterbody samples were examined using grazing-incidence X-ray diffraction with the Co-K α radiation and a beam incidence angle of 5°. A DRON-7 X-ray diffractometer (Burevestnik, Russia) with a scan range of 15–80° and a (2θ) step size of 0.05° was used. The XRD peak identification was carried out with the use of Crystal Impact’s software “Match!” (Version 3.9, Crystal Impact, Bonn, Germany).

The microstructure on the worn surfaces was studied using an Olympus OLS LEXT 4100 laser scanning microscope (Olympus Corp., Tokyo, Japan) and scanning electron microscope (SEM) TESCAN VEGA 3 SBU (TESCAN ORSAY HOLDING, Brno, Czech Republic) equipped with electron energy dispersive spectroscopes (EDS) OXFORD X-Max 50 (Oxford Instruments, Concord, MA, USA) operated at 20 kV, 4–12 nA, and a ~2 µm probe spot size.

## 3. Results

### 3.1. Microstructure of As-Clad Coatings

The as-clad coating was structurally composed of eutectic fishbone-type carbides with fine precipitates and grey matrix grains (Figure 2). According to both EDS (Figure 2d) and XRD (Figure 2e), the reinforcing phases are represented by carbides such as M_6_C, M_2_C, and FeW_3_C. These reinforcing phases have been embedded in the γ-Fe+α-Fe matrix (Figure 2c–e). Apart from that, there were M_2_C carbide agglomerates with the intercarbide spaces occupied by the matrix material. Such a microstructure was similar to that of the metal-ceramic composites (MCC) (Figure 2c,d). The mean microhardness of as-clad coating measured at the load of 100 g was 7900 ± 100 MPa.

### 3.2. Sliding Friction and Wear

The tribological testing allowed for the determination of the dependencies of both the wear rate and coefficient of friction (CoF) on the sliding speed (Figure 3), which characterize the tribological behavior of the coatings as adaptive and demonstrate that both wear rate (Figure 3a) and CoF (Figure 3b) decreased when increasing the sliding speed.

### 3.3. XRD of Worn Surfaces

The glancing X-ray diffractograms with the incidence angle of 5° were obtained from worn surfaces of the M2/W coatings tested at different sliding speeds (Figure 4). It could be observed that sliding at 0.8 m/s and 1.2 m/s had only some quantitative effect on the phase contents, so that the α-Fe and γ-Fe contents decreased and increased, respectively, with the sliding speed. In addition, the intensity of the M_6_*C* carbide XRD peaks grows with speed.

More changes can be observed on sliding at speeds of 2.4 and 3.6 m/s when, along with the increasing heights of their M_6_*C* carbide peaks, there appear new phases such as M_7_C_3_ carbide and mixed FeWO_4_, Fe_2_WO_8_ oxides. Furthermore, in contrast to sliding at low speeds, the content of α-Fe increased with corresponding reduction in the γ-Fe one.

The worn surface of the AISI 52,100 counterbody after sliding at 3.6 m/s demonstrated the presence of both FeO and FeWO_4_ along with the initially detected α-Fe and γ-Fe grains (Figure 5).

### 3.4. Morphology of the Worn Surfaces

The metallographic images in Figure 6a–d demonstrate the worn surfaces of the coating obtained on sliding at different sliding speeds. Sliding at both 0.8 m/s and 1.2 m/s resulted in a regular pattern consisting of thin wear grooves aligned with the sliding speed direction. The higher speed sliding surfaces are characterized by the deterioration of the regular groove pattern as well as by the presence of irregularly shaped islet-type 10–40 μm formations. The worn surface roughness increases with the sliding speed Figure 6a’–d’ and Figure 7).

The roughness of the AISI 52,100 counterbody worn surface obtained on sliding at 3.6 m/s is as follows: R_a_ = 2.47 µm and R_q_ = 0.59 µm. Furthermore, this worn surface allows for the observation of wear grooves and deteriorated zones (Figure 8).

SEM studies show that samples tested at 0.8 and 1.2 м/*c* allow directly observing the initial fishbone carbide microstructure of the as-deposited coating so that neither the mechanically mixed layer (MML) nor any transferred material can be detected (Figure 9 and Figure 10, Table 2 and Table 3). However, there are dark areas that, judging by their EDS spectra, contain much oxygen and iron and, therefore, can be identified as iron oxides (Figure 10b–d; Table 3; 7S–9S).

The areas occupied by carbide agglomerates still contain bright zones enriched with W (Figure 9d–f; Table 3; 13S–15S) and grey zones additionally containing Mo, Fe, Cr, and V (Figure 9d–f; Table 3; 10S, 12S). All these EDS spectra show only minor concentrations of oxygen *(*Table 2 and Table 3).

Sliding at 2.4 m/s allowed obtaining the worn surfaces almost fully covered by the transferred and smeared material (MML) of sufficiently different morphology with the dispersed bright carbide fragments (Figure 11). The carbide agglomerates are still present, whereas no fishbone carbide structures are observed. The dark-grey areas contain much oxygen and metals transferred from the counterbody (Figure 11f, Table 4; 4S–6S, 8S and 9S). The neighboring grey areas, however, are not oxidized (Figure 11f, Table 4; 1S–3S).

Sliding at 3.6 m/s produced the worn surfaces containing dark, irregular-shaped flake-like areas (Figure 12 Table 5; 6S–9S) containing much Fe and O. Furthermore, the neighboring grey ones contain practically inoxidized counterbody steel components (Figure 12, Table 5; 1S, 11S, and 12S). The large areas are occupied by the composite structures with bright carbide fragments dispersed in the grey matrix (Figure 12, Table 5; 1A–3A) that contains much oxygen, tungsten, and iron. It is suggested here that these areas are made of mixed FeWO_4_ and Fe_2_WO_8_ oxides.

The AISI 52,100 counterbody’s worn surface after sliding at 3.6 m/s allows observing three main structural components, such as dark flake-like oxygen- and iron-rich islets (Figure 13, Table 6; 9S–13S), as well as grey and light-grey areas.

### 3.5. Subsurface Microstructures

The subsurface SEM images of microstructures below the worn surfaces of samples tested at 0.8 and 1.2 m/s allow observing brittle fracture of the fishbone carbide structures without any signs of MML (Figure 14).

The subsurface structures obtained after testing at a sliding speed of 2.4 m/s reveal the carbide fragments aligned along the matrix plastic flow direction in accordance with the friction force action (Figure 15a,b,e). The plastic deformation penetration is about 20 μm. The thickness of MML is about 1 μm (Figure 15c,d).

Sliding at 3.6 m/s resulted in a dramatic subsurface structure evolution with the generation of 3–5 μm of thickness MML (Figure 16a,c). The MMLs are composed of bright carbide, heavily oxidized dark areas enriched with O, Fe, W, and Mo (Figure 16d–f; Table 7; 4A and 5A) and light-grey inoxidized ones enriched with Fe, W, and Mo (Figure 16d–f; Table 7; 1A and 2A).

In addition, some continuous Fe-enriched regions can be observed under the MML due to the gray color contrast gradations (Figure 16a,c). The gray regions are enriched with Cr (Figure 16b,d–f; Table 7; 7S–9S) while those of light gray color contain more W (Figure 16b,d–f; Table 7; 10S and 12S) and almost no oxygen. It should be noted that these Cr-enriched regions contain more chromium compared with that of AIS 52,100 steel.

Let us note also that some additional structural components in the form of irregularly shaped particles containing both Cr and W (Figure 16; Table 7; 13S, 15S, and 17S) show up and are darker than those of only W-containing fishbone carbide structure fragments (Figure 16; Table 7; 4S–6S).

The subsurface area on the AISI 52,100 counterbody after sliding at 3.6 m/s demonstrates the presence of 1–1.5 of thickness MML enriched with Fe, O, and W (Figure 17; Table 8; 2A–4A). According to the corresponding XRD pattern (Figure 4), it should be composed of both FeO and FeWO_4_. As a rule, the interface between MML and underlying material is defectless (Figure 17a,b,e,f) but some amount of cracks and fragments are still observable (Figure 17c,d). The MML are also enriched with Cr compared with that in AISI 5210 steel so that some transfer from the M2/WC coating to the AISI 52,100 counterbody can be suggested (Figure 17b–d; Table 8; 1A–4A, 1S and 2S).

## 4. Discussion

### 4.1. The As-Clad Coating Microstructure

Dissolution of WC with the following precipitation of the η-type M_6_C and M_12_C carbides is a typical phenomenon in laser hot-wire deposition and spark plasma sintering of W–Me–C coatings, where M stands for metals Co, Fe, Ni, Ti, Cr, Mn, V, etc. [24,29,30,53]. These η-carbides are brittle and, therefore, detrimental for the material toughness. On the other hand, the metal matrix composites consolidated using the nanosized (Fe,W)_6_C and (Fe,W)_12_C η-carbides as well as fine Fe_3_W_3_C dispersed in a ductile austenitic matrix [30,53] demonstrate high hardness and abrasive wear resistance, therefore, these η-carbides may be used for hardening the coatings. Their hardness is comparable with that of WC, while their cost is sufficiently lower because of their lower tungsten content.

The results obtained in the course of this work as well as those obtained from the literature source about solid state reactions [24,29,30,53] allow proposing a scheme disclosing the phase formation in solidification of the Fe-W-C melt created by electron beam irradiation of the powder bed. The first stage is when hexagonal WC is partially transformed into a mixed and intermediate FeW_3_C carbide:3 WC + Fe → FeW_3_C + 2C↑(1)

According to the corresponding triple phase diagram section at 1250 °C [54], this carbide may be transformed into η-phases, in particular, those may be η_1_-Fe_3_W_3_C and η_2_-Fe_6_W_6_C carbides:3FeW_3_C + 6Fe → Fe_3_W_3_C + Fe_6_W_6_C + C(2)

The resulting structures contain M_6_C (where M stands for Fe, W, V, and Mo) and M_2_C carbides (Mo, W, V, and Cr).

Considering the Fe-W-C triple phase diagram the observed structures and phases were anticipated and described in a number of works devoted to iron-base WC-containing composite coatings obtained using electron beam cladding [8,24] and laser cladding [31,34,36,38].

### 4.2. Wear Transition

Increasing the sliding speed is accompanied by increasing the tribological contact temperature, in particular, the flash temperature of the composite worn surfaces can be estimated using the well-known equation [55]:(3)$$\Delta T=\frac{\mu PV}{4J\left(K_{sample}+K_{disk}\right)\alpha }$$
where *μ* is the CoF, *P* and *V* are the normal load and sliding speed, respectively, and *K_sample_* and *K_disk_* are the thermal conductivities of the pin and disk, respectively. *J* is the Joule’s constant (in this case, *J* = 1) and α is the contact radius of the real contact area to be determined from an equation as follows (4):(4)$$\alpha ={\left(\frac{P}{\pi H_{sample}}\right)}^{1/2\;}$$

$H_{sample}$ is the sample’s hardness.

This flash temperature estimation was carried for the test parameters as follows: *P* = 100 N, hardness 8 GPa, V = 0.8–3.6 m/s and µ = 0.45 − 1 depending on the sliding speed (Figure 3b). Thermal conductivity values of the composite coating and AISI 52,100 were in the ranges 50–80 W⋅m^−1^ K^−1^ and 44–50 W⋅m^−1^ K^−1^ and were borrowed from the literature sources [56,57,58] and [59,60], respectively. The calculated flash temperatures were increasing with the sliding speed in the range 60–290 °C (Figure 18).

Abrasive ploughing was the main wear mechanism in sliding at 0.8 m/s and 1.2 m/s speeds (Figure 6a,a’,b,b’), which was characterized by intensive breaking of the fishbone carbide structures (Figure 14 and Figure 15) as well as high wear rates and CoF values (Figure 3). The XRD results showed that no new phases have been formed on the worn surfaces of coatings (Figure 2 and Figure 4), while only some traces of tribochemical reactions in the form of dark areas enriched with both iron and oxygen were suspected (Figure 9 and Figure 10; Table 3; 7S–9S). These results observed at 0.8 m/s and 1.2 m/s looked similar to those obtained on iron-base WC-reinforced coatings after low-speed sliding [25,35,38,39,40,44,45,46] with a dominant abrasion wear mechanism.

At sliding speeds of 2.4 m/s and 3.6 m/s, the quasi-viscous flow of MML (Figure 15 and Figure 16) that contained both FeWO_4_ and Fe_2_WO_6_ (Figure 4) allowed reducing CoF and wear rate of the M2/W coating in comparison with those obtained by abrasive ploughing at low sliding speeds 0.8 m/s and 1.2 m/s (Figure 3). In addition, on sliding, some amount of M_7_C_3_ (Cr, V, Mo, Fe) has been found below the worn surface.

Increasing the sliding speed to 2.4 m/s and 3.6 m/s causes heating and intensification of the metal transfer from the coating to the counterbody and back, as well as selective oxidizing of the transfer metal components.
2Fe + O_2_ = 2FeO_4_
(5)
FeO + 3O_2_→2Fe_2_O_3_
(6)

Such a pulling out of the matrix results in easier fragmentation of the bone-type carbide structures. The subsurface fracture of the W_2_C, Fe_3_W_3_C, and FeW_3_C grains gave fine fragments that could be easily oxidized into WO_3_ at 800 °C [61]. The next stage might have been the synthesis of self-lubricating iron tungstate according to reactions [62,63]:FeO + WO_3_ = FeWO_4_
(7)
Fe_2_O_3_ + WO_3_ → Fe_2_WO_6_
(8)

It is worthwhile to note that dramatic phase changes occurred in the subsurface of the tested samples with the generation of quasi-viscous tribological layers at relatively low sliding speeds and flash temperatures (Figure 18). Approximately the same structural evolution was observed on WC/Y–TZP–Al_2_O_3_ hybrid ceramic–matrix composites with dispersed Hadfield steel particles after sliding at 37 m/s when the flash temperatures were calculated in the range 726–2110 °C depending upon the WC- content [9].

### 4.3. Subsurface Multi-Layer Structure of Tribooxidized M2/W Coating That Provided the Self-Lubricating Effect and Self-Healing Effect in High-Speed Sliding

Let us note that the presence of different metals in the MML, which give different levels of brightness in the corresponding SEM BSE images, served to better identify the quasi-viscous flow of these layers (Figure 16a,c). It is interesting that such a flow was in continuous mode and did not form any discontinuities, while the self-healing capacity was revealed when damaged areas and cracks were filled with a quasi-viscous mass. This type of behavior was similar to that found in the high-speed sliding of a hybrid WC/Y–TZP–Al_2_O_3_ composite with dispersed Hadfield steel particles [9], where higher wear resistance and simultaneously reduced friction were achieved owing to the quasi-viscous flow of the MML impregnated with the in-situ synthesized FeWO_4_ and Fe_2_WO_6_.

## 5. Conclusions

Tribological testing of the electron beam-clad composite coatings obtained from a mixture of powdered M2 steel and WC has been carried out in contact with a AISI 52,100 counterbody. The coating structure consisted of an iron-base γ+α’ matrix reinforced by fishbone carbide structures. Simultaneous friction and wear rate reduction in the sliding speed range of 0.8–3.6 m/s was obtained on the M2/W coatings. The study of the worn surfaces of the coatings showed that the wear mechanism transition occurred when sliding the M2/W coating against the AISI 52,100 disk in the sliding speed range of 0.8 to 3.6 m/s. This transition was attributed to the in-situ tribochemical generation of FeWO_4_ and Fe_2_WO_6_ at sliding speeds of 2.4 and 3.6 m/s, which then allowed the generation of a mechanically mixed layer compacted of the said tungstates, carbide fragments, and metallic wear debris. These subsurface MML structures provided the lubrication effect as well as protected the underlying coating structures against abrasion and adhesion. The quasi-viscous flow of these layers was observed.

The specificity of such an adaptation mechanism is that it was realized at relatively low sliding speeds, which corresponded to a maximum calculated flash temperature of about 290 °C.

## Figures and Tables

**Figure 1 materials-16-01013-f001:**
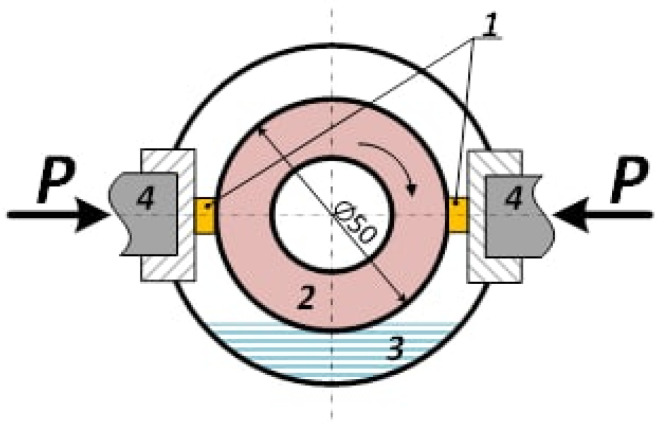
Tribological test scheme. 1—blocks, 2—disk, 3—water, 4—pressure hydraulic cylinders.

**Figure 2 materials-16-01013-f002:**
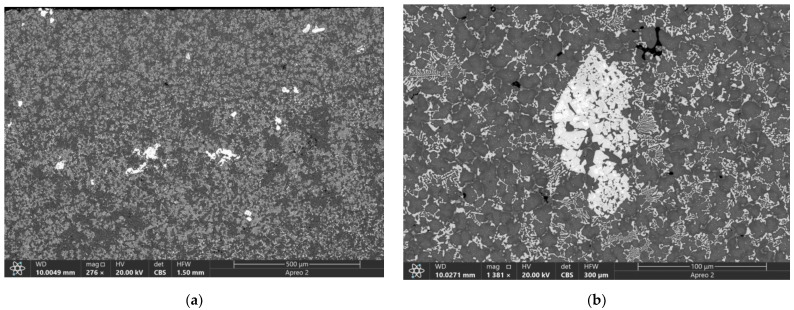
The SEM BSE images (**a**–**c**) and overlay EDS image (**d**) of the as-clad M2/W coating transverse section. Glancing X-ray diffractograms of initial surface of M2/W coatings (**e**).

**Figure 3 materials-16-01013-f003:**
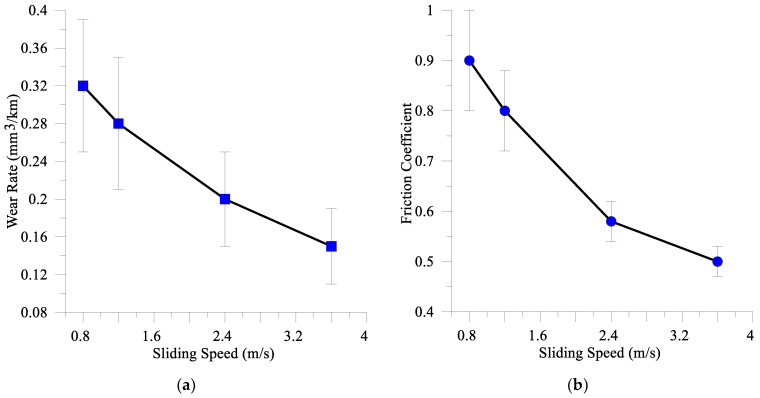
Wear rate (**a**) and coefficient of friction (**b**) vs. sliding speed for M2/W coating rubbed against the AISI 52,100 counterbody.

**Figure 4 materials-16-01013-f004:**
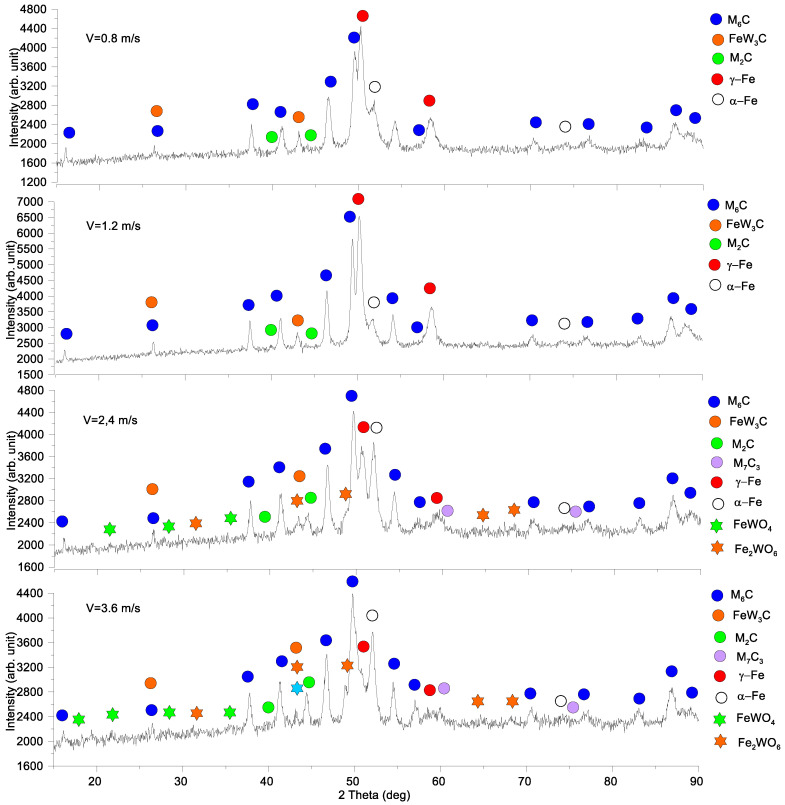
Glancing X-ray diffractograms of worn surfaces of M2/W coatings.

**Figure 5 materials-16-01013-f005:**
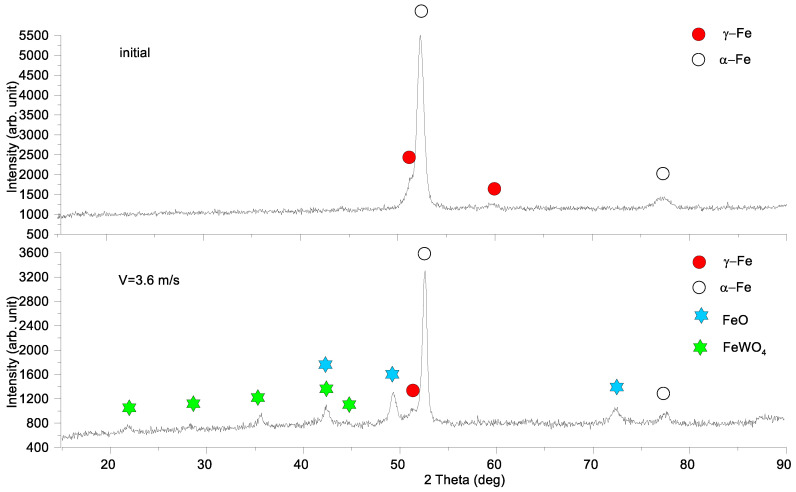
Glancing X-ray diffractograms of initial and worn surface of AISI 52,100 counterbody with incidence angle 5°. Data presented for specimens after sliding speed of 3.6 m/s.

**Figure 6 materials-16-01013-f006:**
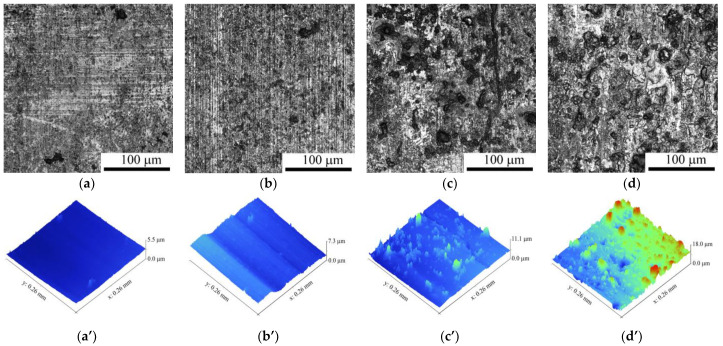
The optical images of worn surface (**a**–**d**) and 3D maps of the worn surface (**a’**–**d’**) of of M2/W coating after sliding speeds of: 0.8 m/s (**a**,**a’**); 1.2 m/s (**b**,**b’**); 2.4 m/s (**c**,**c’**); 3.6 m/s (**d**,**d’**).

**Figure 7 materials-16-01013-f007:**
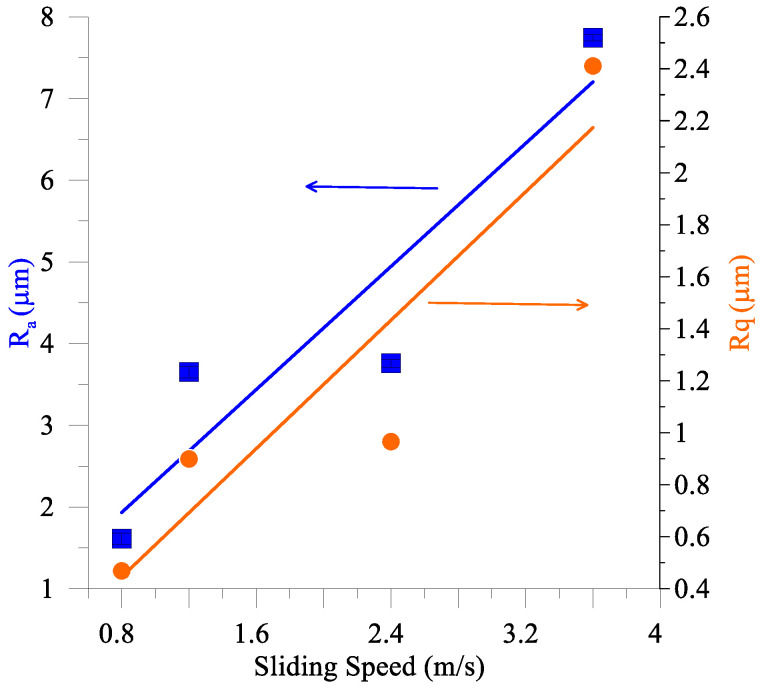
Arithmetical mean and root mean square deviations of worn surface roughnesses, Ra and Rq, resp of M2/W coating after different sliding speeds.

**Figure 8 materials-16-01013-f008:**
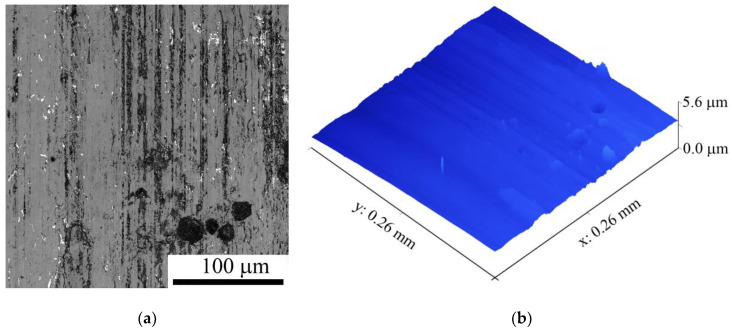
The optical images worn surface (**a**) and 3D map of the worn surface (**b**) of AISI 52,100 counterbody after sliding speed of 3.6 m/s.

**Figure 9 materials-16-01013-f009:**
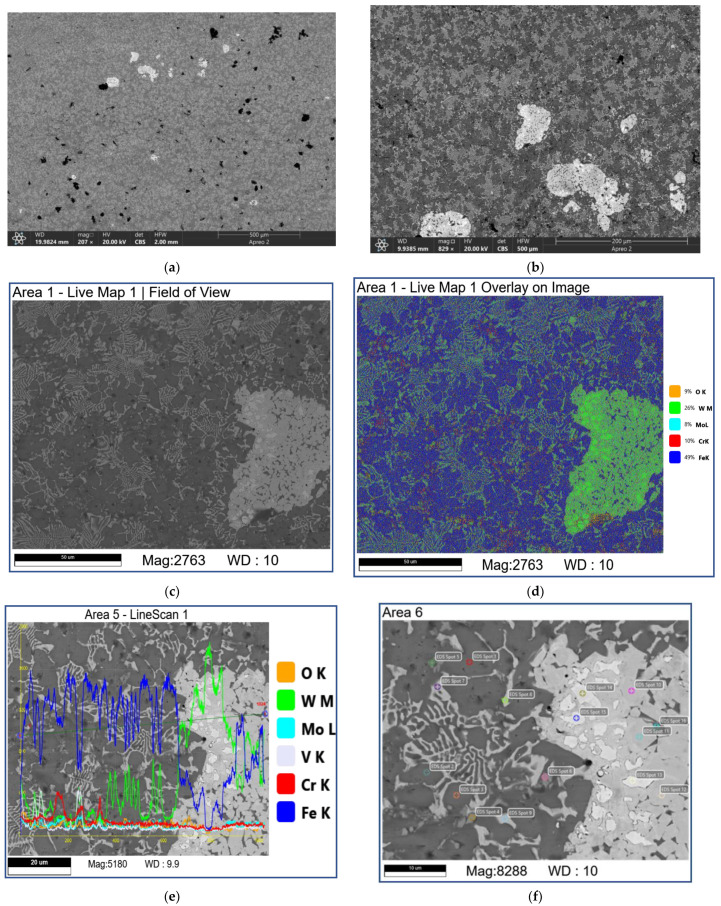
The SEM BSE images (**a**–**c**,**e**,**f**), overlay EDS image (**d**) and EDS line scans (**e**) of M2/W coating worn surface. Data presented for specimens after sliding speed of 0.8 m/s. Numbers on (**f**) indicate probe zones for which EDS elemental concentrations were determined indicated in Table 2.

**Figure 10 materials-16-01013-f010:**
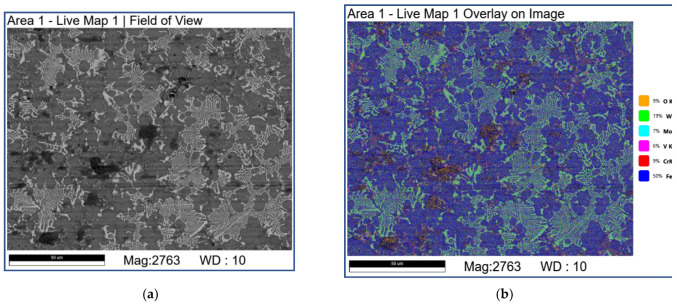
The SEM BSE images (**a**,**c**,**d**), overlay EDS image (**b**) and EDS line scans (**c**) of M2/W coating worn surface. Data presented for specimens after sliding speed of 1.2 m/s. Numbers on (**d**) indicate probe zones for which EDS elemental concentrations were determined indicated in Table 3.

**Figure 11 materials-16-01013-f011:**
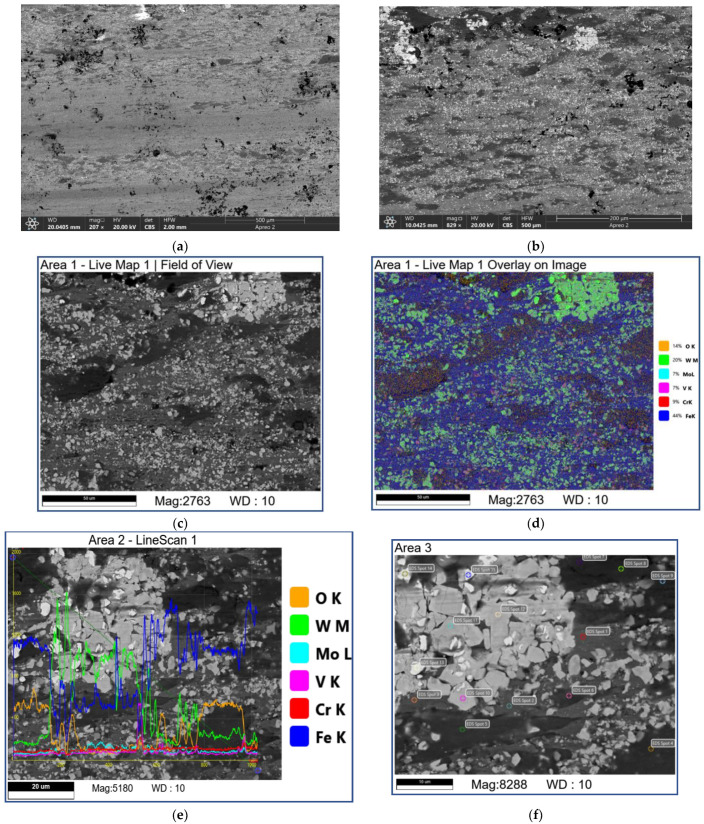
The SEM BSE images (**a**–**c**,**e**,**f**), overlay EDS image (**d**) and EDS line scans (**e**) of M2/W coating worn surface. Data presented for specimens after sliding speed of 2.4 m/s. Numbers on (**f**) indicate probe zones for which EDS elemental concentrations were determined indicated in Table 4.

**Figure 12 materials-16-01013-f012:**
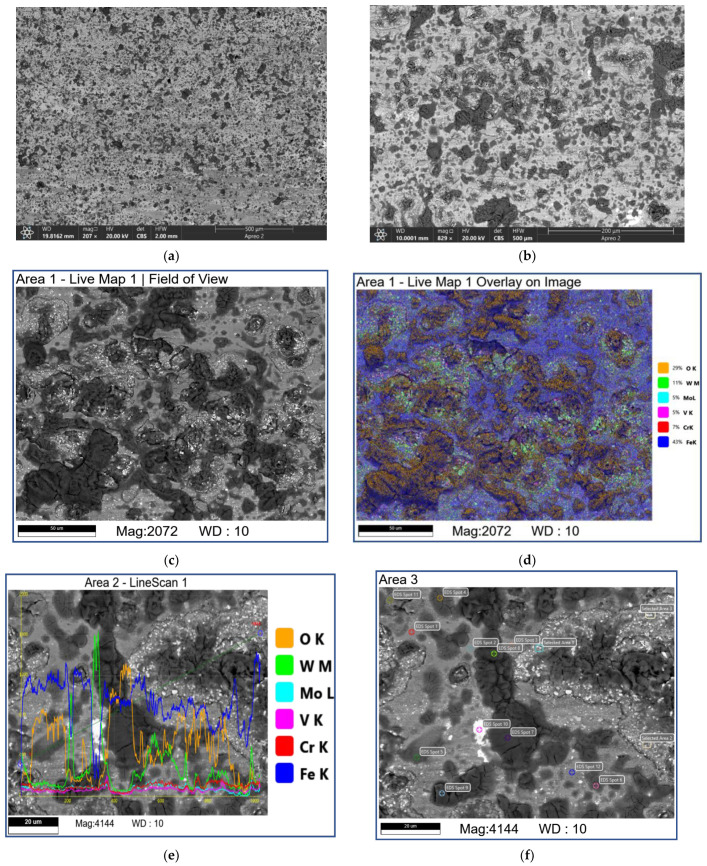
The SEM BSE images (**a**–**c**,**e**,**f**), overlay EDS image (**d**) and EDS line scans (**e**) of M2/W coating worn surface. Data presented for specimens after sliding speed of 3.6 m/s. Numbers on (**f**) indicate probe zones for which EDS elemental concentrations were determined indicated in Table 5.

**Figure 13 materials-16-01013-f013:**
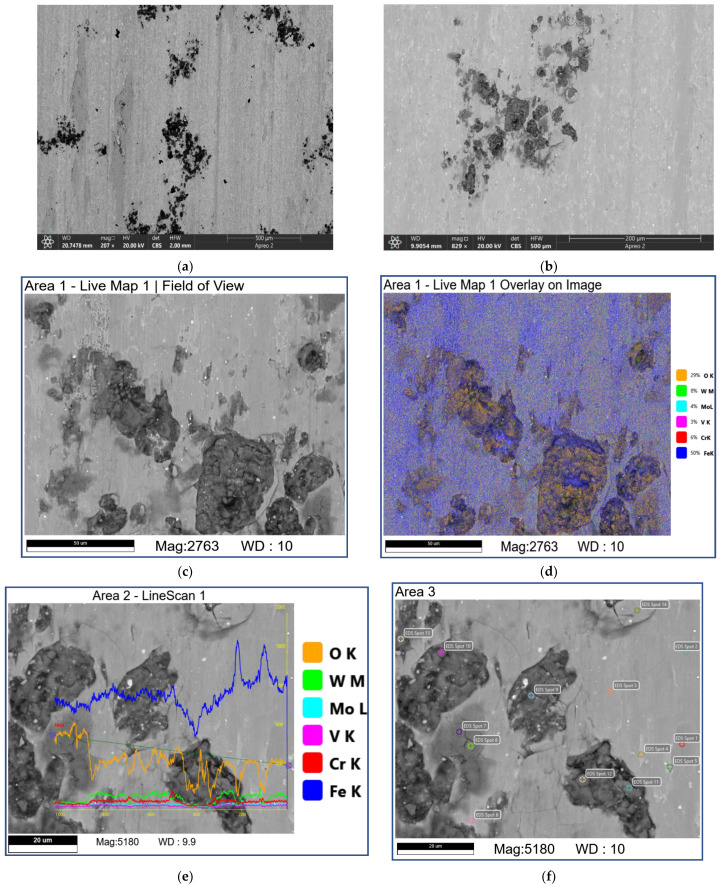
The SEM BSE images (**a**–**c**,**e**,**f**), overlay EDS image (**d**) and EDS line scans (**e**) of AISI 52,100 counterbody worn surface. Data presented for specimens after sliding speed of 3.6 m/s. Numbers on (**f**) indicate probe zones for which EDS elemental concentrations were determined indicated in Table 6.

**Figure 14 materials-16-01013-f014:**
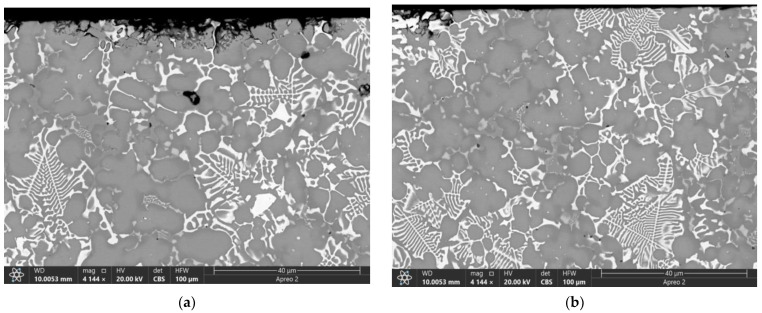
The SEM BSE images (**a**–**c**,**e**) and overlay EDS images (**d**,**f**) of the M2/W coating subsurface microstructures. Data presented for specimens after sliding at 0.8 m/s.

**Figure 15 materials-16-01013-f015:**
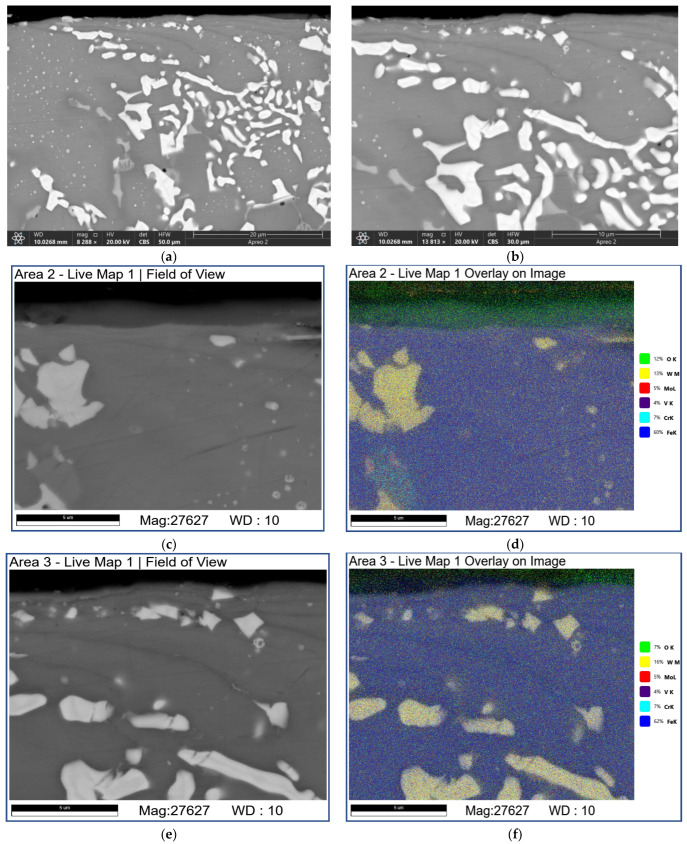
The SEM BSE images (**a**–**c**,**e**) and overlay EDS images (**d**,**f**) of M2/W coating subsurface microstructures below the worn surface. Data presented for specimens after sliding speed of 2.4 m/s.

**Figure 16 materials-16-01013-f016:**
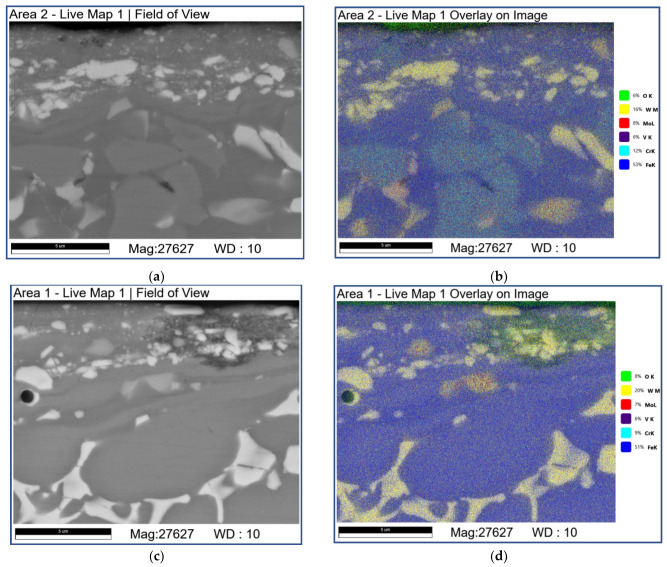
The SEM BSE images (**a**–**c**,**e**,**f**), overlay EDS images (**b**,**d**) and EDS line scans (**e**) of the subsurface M2/W coating microstructures formed by sliding. Data presented for specimens on sliding speed of 3.6 m/s. Numbers on (**f**) indicate probe zones for which EDS elemental concentrations were determined indicated in Table 7.

**Figure 17 materials-16-01013-f017:**
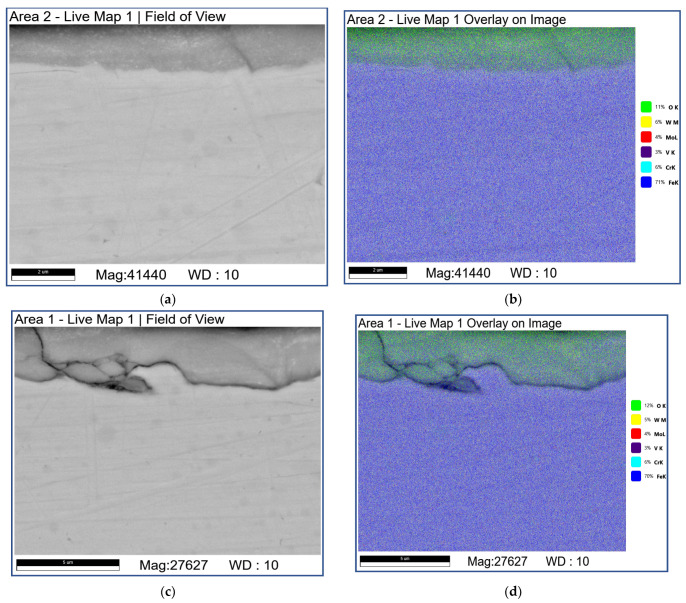
The SEM BSE images (**a**,**c**,**e**,**f**), overlay EDS image (**b**,**d**) and EDS line scans (**e**) of AISI 52,100 subsurface structures formed by sliding. Data presented for specimens after sliding speed of 3.6 m/s. Numbers on (**f**) indicate probe zones for which EDS elemental concentrations were determined indicated in Table 8.

**Figure 18 materials-16-01013-f018:**
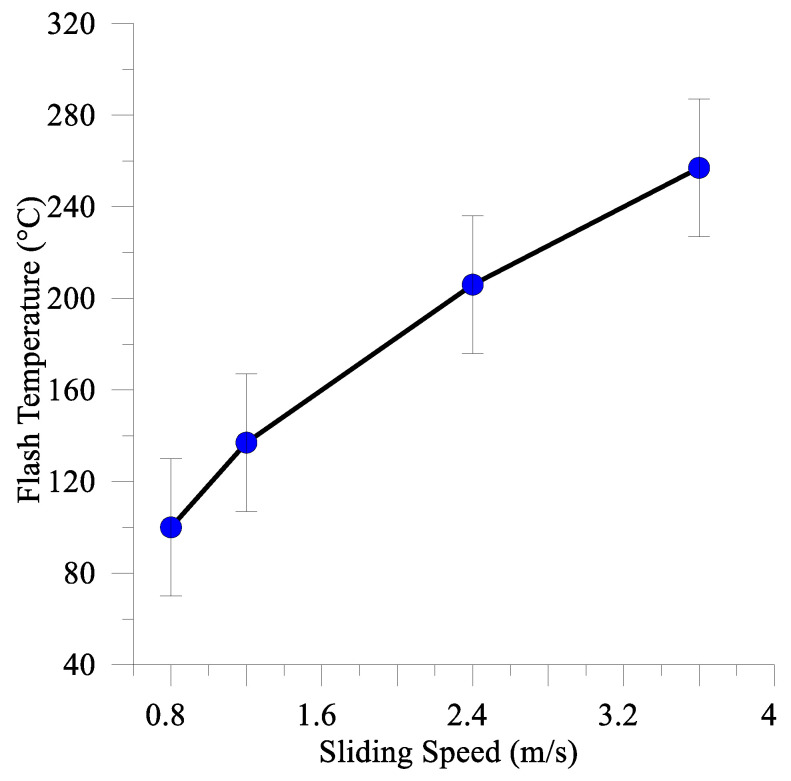
Flash temperature vs. sliding speed for M2/W coating rubbed against the AISI 52,100 counterbody.

**Table 1 materials-16-01013-t001:** Chemical composition of the counterbody steel.

Element, wt.%								
C	Si	Mn	Cr	Ni	S	P	Cu	Fe
0.95–1.05	0.17–0.37	0.20–0.40	1.30–1.65	≤0.30	≤0.02	≤0.027	≤0.25	Bal.

**Table 2 materials-16-01013-t002:** The EDS elemental composition of coating M2/W worn surface after sliding at 0.8 m/s (Figure 9f).

Area	Element, wt.% (at.%)					
	O	V	Cr	Fe	Mo	W
1S	3.3 (11.5)	1.2 (1.3)	3.4 (3.6)	81.8 (80.3)	1.2 (0.7)	9.1 (2.7)
2S	1.0 (3.5)	1.0 (1.1)	3.6 (4.0)	85.7 (88.3)	1.1 (0.7)	7.7 (2.4)
3S	0.8 (3.0)	1.0 (1.2)	3.6 (4.0)	85.8 (88.8)	1.0 (0.6)	7.7 (2.4)
4S	8.0 (26.9)	15.4 (16.4)	5.5 (5.7)	42.1 (40.9)	5.5 (3.1)	23.5 (6.9)
5S	8.7 (28.1)	12.6 (12.8)	6.6 (6.6)	47.9 (44.4)	4.8 (2.6)	19.4 (5.5)
6S	9.9 (34.6)	27.9 (30.6)	4.4 (4.8)	14.8 (14.8)	8.0 (4.6)	35.0 (10.6)
7S	2.8 (12.7)	2.8 (4.0)	3.9 (5.5)	43.5 (57.0)	5.8 (4.4)	41.2 (16.4)
8S	1.5 (6.0)	1.9 (2.3)	4.3 (5.1)	71.0 (78.8)	2.1 (1.3)	19.3 (6.5)
9S	4.1 (18.1)	2.6 (3.6)	3.2 (4.3)	42.6 (53.7)	6.1 (4.5)	41.5 (15.9)
10S	1.2 (7.1)	2.1 (3.7)	3.3 (5.9)	29.7 (49.1)	4.8 (4.6)	58.9 (29.6)
11S	1.3 (7.5)	2.0 (3.7)	3.3 (5.8)	29.9 (49.1)	4.8 (4.6)	58.7 (29.3)
12S	1.2 (7.0)	1.9 (3.4)	3.1 (5.7)	28.9 (48.6)	4.5 (4.4)	60.4 (30.9)
13S	3.3 (27.1)	0.0 (0.1)	0.1 (0.2)	2.6 (6.0)	0.2 (0.3)	93.8 (66.4)
14S	1.2 (10.9)	0.2 (0.7)	0.3 (0.9)	3.8 (10.1)	0.1 (0.1)	94.5 (77.3)
15S	4.8 (28.9)	0.6 (1.1)	1.2 (2.1)	16.2 (27.7)	0.3 (0.3)	76.9 (39.9)
16S	2.1 (9.1)	1.4 (1.9)	3.2 (4.3)	55.5 (69.4)	2.8 (2.1)	35.1 (13.3)

**Table 3 materials-16-01013-t003:** EDS elemental composition of the M2/W worn surface on sliding at 1.2 m/s (Figure 10d).

Area	Element, wt.% (at.%)					
	O	V	Cr	Fe	Mo	W
1S	1.0 (3.6)	1.4 (1.6)	4.2 (4.6)	84.6 (87.0)	1.5 (0.9)	7.3 (2.3)
2S	1.7 (5.9)	1.2 (1.3)	3.8 (4.1)	85.4 (86.1)	0.9 (0.5)	7.2 (2.2)
3S	1.7 (6.3)	1.0 (1.2)	3.3 (3.6)	82.6 (85.0)	1.3 (0.8)	10.1 (3.2)
4S	2.0 (7.0)	2.0 (2.2)	15.3 (16.6)	68.9 (69.4)	4.5 (2.7)	7.2 (2.2)
5S	3.0 (10.2)	1.9 (2.0)	14.8 (15.6)	68.8 (67.5)	4.3 (2.5)	7.3 (2.2)
6S	1.9 (6.7)	1.0 (1.1)	3.6 (3.9)	84.8 (85.3)	1.1 (0.6)	7.6 (2.3)
7S	24.7 (54.5)	0.8 (0.5)	2.5 (1.7)	66.6 (42.1)	0.6 (0.2)	4.8 (0.9)
8S	27.4 (58.1)	0.7 (0.5)	2.2 (1.4)	63.6 (38.7)	0.8 (0.3)	5.3 (1.0)
9S	22.7 (51.8)	0.6 (0.4)	4.0 (2.8)	66.8 (43.6)	1.1 (0.4)	4.8 (0.9)
10S	19.5 (46.7)	0.8 (0.6)	3.1 (2.3)	71.7 (49.2)	0.6 (0.3)	4.3 (0.9)
11S	22.5 (51.5)	0.8 (0.6)	2.8 (2.0)	68.3 (44.7)	0.8 (0.3)	4.8 (0.9)
12S	1.9 (9.6)	3.0 (4.9)	3.8 (6.0)	35.1 (51.9)	5.8 (5.0)	50.5 (22.7)
13S	2.3 (11.2)	3.2 (5.0)	4.0 (6.0)	37.1 (52.6)	5.4 (4.4)	48.1 (20.7)
14S	1.8 (9.0)	2.6 (4.2)	4.4 (6.8)	37.2 (53.4)	7.5 (6.2)	46.5 (20.3)

**Table 4 materials-16-01013-t004:** EDS elemental composition of the M2/W worn surface after sliding at 2.4 m/s (Figure 11f).

Area	Element, wt.% (at.%)					
	O	V	Cr	Fe	Mo	W
1S	1.2 (4.2)	1.0 (1.1)	3.5 (3.8)	85.7 (87.9)	1.0 (0.6)	7.7 (2.4)
2S	1.1 (3.8)	1.0 (1.1)	3.8 (4.1)	87.1 (88.5)	1.0 (0.6)	6.0 (1.9)
3S	1.4 (5.0)	1.1 (1.2)	3.6 (3.9)	86.6 (87.4)	0.9 (0.6)	6.3 (1.9)
4S	22.4 (52.4)	1.1 (0.8)	3.2 (2.3)	62.9 (42.1)	1.4 (0.6)	8.8 (1.8)
5S	24.2 (55.5)	1.2 (0.9)	3.5 (2.4)	58.3 (38.3)	1.8 (0.7)	11.1 (2.2)
6S	16.9 (42.9)	1.0 (0.8)	3.2 (2.5)	71.6 (52.1)	0.9 (0.4)	6.5 (1.4)
7S	23.9 (53.8)	0.7 (0.5)	2.6 (1.8)	65.4 (42.3)	0.9 (0.4)	6.4 (1.3)
8S	22.4 (51.5)	0.8 (0.6)	2.9 (2.1)	67.6 (44.5)	0.8 (0.3)	5.5 (1.1)
9S	23.1 (52.4)	1.0 (0.7)	2.8 (2.0)	66.9 (43.5)	0.8 (0.3)	5.5 (1.1)
10S	2.9 (15.0)	1.9 (3.1)	3.1 (5.1)	31.3 (47.1)	4.8 (4.2)	56.0 (25.6)
11S	5.6 (26.5)	1.8 (2.7)	2.8 (3.9)	31.6 (41.7)	5.4 (4.2)	52.7 (21.1)
12S	2.3 (12.5)	1.9 (3.3)	3.1 (5.2)	30.2 (47.4)	3.9 (3.6)	58.6 (28.0)
13S	1.1 (10.2)	0.4 (1.1)	0.6 (1.7)	4.6 (12.0)	0.3 (0.5)	93.0 (74.5)
14S	1.3 (12.3)	0.2 (0.5)	0.2 (0.6)	3.1 (8.4)	0.1 (0.1)	95.2 (78.1)
15S	1.3 (11.8)	0.6 (1.7)	1.1 (3.1)	4.9 (12.4)	0.2 (0.3)	91.8 (70.7)

**Table 5 materials-16-01013-t005:** EDS chemical composition of the M2/W worn surface on sliding at 3.6 m/s (Figure 12f).

Area	Element, wt.% (at.%)					
	O	V	Cr	Fe	Mo	W
1S	1.6 (5.6)	0.3 (0.4)	1.3 (0.9)	92.8 (91.0)	1.9 (1.1)	2.1 (0.6)
2S	1.7 (5.9)	0.5 (0.5)	1.5 (1.6)	91.5 (90.0)	2.1 (1.2)	2.7 (0.8)
3S	4.5 (14.6)	0.3 (0.3)	1.3 (1.3)	89.4 (82.1)	1.8 (1.0)	2.6 (0.7)
4S	26.1 (56.1)	0.6 (0.4)	1.9 (1.3)	66.7 (41.1)	1.6 (0.6)	3.1 (0.6)
5S	27.5 (58.1)	0.8 (0.5)	2.0 (1.3)	63.7 (38.6)	2.3 (0.8)	3.8 (0.7)
6S	24.7 (54.2)	0.5 (0.4)	1.9 (1.3)	68.2 (42.9)	1.9 (0.7)	2.8 (0.5)
7S	28.3 (58.5)	0.1 (0.1)	0.5 (0.3)	67.8 (40.2)	2.2 (0.8)	1.1 (0.2)
8S	35.0 (66.2)	0.1 (0.1)	0.5 (0.3)	59.0 (32.0)	3.4 (1.1)	1.9 (0.3)
9S	31.1 (61.7)	0.2 (0.1)	0.7 (0.4)	65.1 (37.0)	1.9 (0.6)	1.1 (0.2)
10S	1.9 (14.2)	1.0 (2.2)	1.3 (2.9)	12.9 (27.1)	0.6 (0.8)	82.4 (52.8)
11S	15.1 (41.2)	0.7 (0.6)	2.7 (2.2)	66.5 (51.8)	3.3 (1.5)	11.7 (2.8)
12S	10.0 (29.0)	0.9 (0.8)	3.0 (2.7)	78.1 (65.0)	1.9 (0.9)	6.1 (1.6)
1A	22.2 (55.9)	2.0 (1.6)	4.5 (3.5)	45.3 (32.7)	3.3 (1.4)	22.6 (4.9)
2A	19.8 (50.9)	2.0 (1.6)	4.3 (3.4)	52.1 (38.5)	3.0 (1.1)	18.8 (4.2)
3A	15.1 (42.8)	2.0 (1.8)	4.6 (4.0)	55.8 (45.2)	2.9 (1.4)	19.5 (4.8)

**Table 6 materials-16-01013-t006:** EDS chemical composition of the AISI 52,100 counterbody worn surface on sliding at 3.6 m/s (Figure 13f).

Area	Element, wt.% (at.%)					
	O	V	Cr	Fe	Mo	W
1S	19.5 (46.6)	0.7 (0.7)	2.9 (2.1)	72.5 (49.7)	0.5 (0.2)	4.0 (0.8)
2S	20.4 (48.8)	0.8 (0.6)	2.7 (2.0)	68.6 (46.9)	0.9 (0.3)	6.7 (1.4)
3S	18.6 (45.8)	0.6 (0.5)	2.7 (2.1)	70.6 (49.9)	0.7 (0.3)	6.7 (1.4)
4S	2.5 (8.2)	0.1 (0.1)	1.7 (1.7)	94.8 (89.7)	0.0 (0.0)	0.9 (0.3)
5S	2.0 (6.6)	0.2 (0.3)	2.1 (2.2)	94.6 (90.6)	0.3 (0.3)	0.8 (0.2)
6S	7.4 (22.0)	0.1 (0.1)	1.8 (1.7)	89.0 (75.8)	0.1 (0.0)	1.6 (0.4)
7S	32.0 (62.7)	0.3 (0.3)	1.4 (0.9)	63.2 (35.6)	1.0 (0.3)	2.1 (0.4)
8S	30.4 (61.2)	0.3 (0.2)	1.5 (0.9)	63.7 (36.7)	1.8 (0.6)	2.2 (0.4)
9S	29.9 (60.4)	0.1 (0.0)	1.7 (1.0)	64.4 (36.6)	2.0 (0.7)	1.4 (0.3)
10S	24.1 (65.5)	0.1 (0.0)	1.1 (0.7)	59.1 (32.5)	2.5 (0.8)	3.1 (0.5)
11S	31.7 (62.4)	0.0 (0.0)	1.1 (0.7)	63.8 (36.0)	2.0 (0.7)	1.2 (0.2)
12S	38.5 (69.1)	0.1 (0.0)	0.7 (0.4)	58.2 (29.9)	1.4 (0.4)	1.2 (0.2)
13S	34.3 (65.6)	0.1 (0.0)	1.1 (1.1)	59.5 (32.6)	1.9 (0.6)	3.1 (0.5)
14S	19.1 (46.5)	0.9 (0.7)	2.9 (2.2)	70.4 (49.1)	0.6 (0.2)	6.1 (1.3)

**Table 7 materials-16-01013-t007:** EDS elemental composition of the subsurface area on coating M2/W after sliding at 3.6 m/s (Figure 16f).

Area	Element, wt.% (at.%)					
	O	V	Cr	Fe	Mo	W
1S	0.8 (2.8)	1.0 (1.1)	3.6 (4.0)	84.6 (88.5)	1.4 (0.8)	8.8 (2.8)
2S	0.7 (2.5)	1.0 (1.2)	3.6 (4.0)	84.2 (88.5)	1.3 (0.8)	9.2 (3.0)
3S	0.9 (3.2)	0.9 (1.1)	3.3 (3.8)	84.0 (88.2)	1.2 (0.7)	9.6 (3.1)
4S	1.0 (5.2)	2.2 (3.6)	2.9 (4.7)	39.8 (59.3)	6.7 (5.8)	47.3 (21.4)
5S	1.1 (5.2)	2.0 (3.0)	3.2 (4.7)	48.7 (66.2)	6.2 (4.9)	38.8 (16.0)
6S	0.9 (5.0)	2.7 (4.6)	3.1 (5.1)	35.6 (55.1)	7.2 (6.5)	50.5 (23.8)
7S	0.7 (2.5)	0.9 (1.0)	4.2 (4.6)	86.8 (89.2)	1.3 (0.8)	6.1 (1.9)
8S	0.8 (3.0)	2.2 (2.5)	4.7 (5.2)	82.6 (85.8)	1.6 (1.0)	8.1 (2.6)
9S	0.5 (1.8)	0.9 (1.1)	4.0 (4.5)	84.4 (89.0)	1.4 (0.9)	8.8 (2.8)
10S	0.7 (2.7)	0.9 (1.1)	3.3 (3.8)	82.6 (88.1)	1.3 (0.8)	11.2 (3.6)
11S	0.8 (3.1)	1.1 (1.3)	3.5 (4.0)	81.7 (87.0)	1.5 (0.9)	11.3 (3.6)
12S	1.4 (5.0)	0.9 (1.0)	3.4 (3.8)	80.2 (85.0)	1.7 (1.1)	12.5 (4.0)
13S	2.7 (11.8)	26.3 (36.0)	7.7 (10.4)	15.5 (19.3)	12.5 (9.1)	35.3 (13.4)
14S	1.6 (6.4)	12.1 (14.8)	4.8 (5.8)	56.3 (63.1)	4.2 (2.8)	20.9 (7.1)
15S	0.0 (0.0)	15.1 (20.9)	7.3 (9.8)	41.8 (52.7)	8.0 (5.9)	27.8 (10.7)
16S	9.4 (33.2)	1.6 (1.8)	3.6 (3.9)	47.0 (47.7)	5.3 (3.1)	33.1 (10.2)
17S	15.4 (45.5)	2.0 (1.8)	5.1 (4.7)	46.3 (39.1)	4.2 (2.1)	27.0 (6.9)
1A	2.7 (9.9)	1.2 (1.4)	3.7 (4.2)	72.4 (77.0)	3.8 (2.4)	16.3 (5.2)
2A	2.5 (9.4)	1.0 (1.2)	3.2 (3.8)	69.9 (76.6)	4.0 (2.6)	19.3 (6.4)
3A	18.2 (48.3)	1.6 (1.4)	5.2 (4.2)	52.9 (40.2)	3.9 (1.7)	18.3 (4.2)
4A	15.2 (45.0)	1.9 (1.8)	5.0 (4.6)	46.6 (39.4)	5.2 (2.6)	26.0 (6.7)
5A	14.4 (42.2)	1.8 (1.7)	5.0 (4.5)	52.4 (44.0)	3.6 (1.8)	22.8 (5.8)

**Table 8 materials-16-01013-t008:** EDS elemental composition of subsurface area on AISI 52,100 counterbody after sliding at 3.6 m/s (Figure 17f).

Area	Element, wt.% (at.%)					
	O	V	Cr	Fe	Mo	W
1S	1.9 (6.5)	0.0 (0.0)	1.5 (1.5)	95.8 (91.8)	0.0 (0.0)	0.7 (0.2)
2S	1.9 (6.2)	0.0 (0.0)	1.7 (1.8)	95.6 (91.7)	0.0 (0.0)	0.7 (0.2)
1A	28.0 (60.9)	0.5 (0.4)	2.1 (1.4)	55.2 (34.4)	1.3 (0.5)	13.0 (2.5)
2A	21.9 (51.2)	0.5 (0.4)	2.5 (1.8)	67.0 (44.9)	0.6 (0.2)	7.4 (1.5)
3A	23.8 (54.7)	0.5 (0.4)	2.3 (1.6)	62.3 (41.0)	0.9 (0.4)	10.1 (2.0)
4A	26.0 (57.2)	0.5 (0.4)	2.3 (1.6)	61.8 (38.9)	0.7 (0.3)	8.6 (1.6)

## Data Availability

Not applicable.

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
