# Peer review of "Evidence of Tribological Adaptation Controlled by Tribosynthesis of FeWO4 on an WC-Reinforced Electron Beam M2 Steel Coating Rubbed against a HSS Disk in a Range of Sliding Speeds"

_materials, 2023, doi:10.3390/ma16031013_

Round 1
Reviewer 1 Report
Referee Report on “Evidence of Tribological Adaptation Controlled by Tribosynthesis of FeWO4 on an WC-reinforced Electron Beam Deposited HSS M2 Steel Rubbed against a HSS Disk in a Range of Sliding Speeds”
This is a very good paper, that could be certainly recommended for publication, but only after some of the improvements formulated below.
1. MWO4 compounds have applications as wide as scintillators and various other detectors. This fact must be reflected in the introductions because it will be of interest to a wider readership and increase the visibility and interest in the work. See , for example:
PbWO4: Millers, D., Grigorjeva, L., Chernov, S., et al (1997). The temperature dependence of scintillation parameters in PbWO4 crystals. physica status solidi (b), 203(2), 585-589.
ZnWO4: Lisitsyn, V. M., Karipbayev, Z. T., Lisitsyna, L. A., et al. (2016). Effects of Doped Oxygen on ZnWO4 Crystal Luminescence. Key Engineering Materials, Vol. 712, pp. 345-350.
NiWO4. Kuzmin, A., Pankratov, V., Kalinko, A., et al. (2016). UV-VUV synchrotron radiation spectroscopy of NiWO4. Low Temperature Physics, 42(7), 543-546.
2. X-ray diffraction alone is not as informative as in combination with other methods. Much more information could have been obtained if the authors had used Raman spectra measurements. Was it real?
3. About porosity. Actually, pore evolution is a difficult problem with its dynamics and unique transformation mechanisms and it is clear that it should be treated with great attention.
4. Conclusion is quite short. In the new version of the conclusion, please clearly state what new data on studied materials were obtained in this work.
In principle, the article is interesting and can be recommended for publication after due consideration of the above comments.
Author Response
- MWO4 compounds have applications as wide as scintillators and various other detectors. This fact must be reflected in the introductions because it will be of interest to a wider readership and increase the visibility and interest in the work. See , for example:
PbWO4: Millers, D., Grigorjeva, L., Chernov, S., et al (1997). The temperature dependence of scintillation parameters in PbWO4 crystals. physica status solidi (b), 203(2), 585-589.
ZnWO4: Lisitsyn, V. M., Karipbayev, Z. T., Lisitsyna, L. A., et al. (2016). Effects of Doped Oxygen on ZnWO4 Crystal Luminescence. Key Engineering Materials, Vol. 712, pp. 345-350.
NiWO4. Kuzmin, A., Pankratov, V., Kalinko, A., et al. (2016). UV-VUV synchrotron radiation spectroscopy of NiWO4. Low Temperature Physics, 42(7), 543-546.
A: Thank you. These references as well as some other ones have been added to the Introduction section.
- X-ray diffraction alone is not as informative as in combination with other methods. Much more information could have been obtained if the authors had used Raman spectra measurements. Was it real?
A: Thank you. The grazing incidence X-ray diffraction (GIAXD) method was used in combination with the EDS maps, profiles and point probe zones and therefore allowed correct detecting phases formed in tribologically affected layers below the worn surfaces.
We agree that using as much as possible methods in combinations allows obtaining more interesting and reliable results. However, our intentions often do not meet our opportunities.
- About porosity. Actually, pore evolution is a difficult problem with its dynamics and unique transformation mechanisms and it is clear that it should be treated with great attention.
A: Added
- Conclusion is quite short. In the new version of the conclusion, please clearly state what new data on studied materials were obtained in this work.
A: Rewritten

Reviewer 2 Report
Results are quite interesting and manuscript is in relatively good shape. However the following point need to be address:
[1]. Abstract needs to be rewritten again based on some standardized format that include the strategy, execution and obtained results trend.
[2]. Keywords are not appropriate that must be specific words related to your research.
[3]. Technically there is a missing paragraph in introduction that highlights the Tribosynthesis, the other method of cladding deposition or the other reference of electron beam deposition for this or similar materials.
[4]. The last of introduction that highlights the significance and novelty of work is quite weak.
[5]. Remove the unnecessary things from Fig. 1(a) like chair, and include the close and valuable picture.
[6]. If possible unite the separate sections of results and discussion.
[7]. XRD results are included in Section 3.1., so it not good to claim the existence of carbide phase before discussing or presenting the results of XRD.
[8]. Why the legend in Figures 4 is tilting, is it a scanned image or plotted?
[9]. Is it the right heading for Section 3.4? These are microstructures or macro surface images?
[10].The values written in Fig. 6(a’, b’, c’, d’) are blur, and how you plotted these roughness plot?
[11].Conclusion section is completely vague, which needs to be rearranged and rewritten.
Author Response
[1]. Abstract needs to be rewritten again based on some standardized format that include the strategy, execution and obtained results trend.
A: Revised
[2]. Keywords are not appropriate that must be specific words related to your research.
A: Revised
[3]. Technically there is a missing paragraph in introduction that highlights the Tribosynthesis, the other method of cladding deposition or the other reference of electron beam deposition for this or similar materials.
A: The missing information has been added
[4]. The last of introduction that highlights the significance and novelty of work is quite weak.
[5]. Remove the unnecessary things from Fig. 1(a) like chair, and include the close and valuable picture.
A: Thank you. Removed
[6]. If possible unite the separate sections of results and discussion.
A: Thank you. It is obvious for us that the Discussion section should be extended for more detailed discussing the results obtained and relating them with the known ones to elaborate some conception. Also we believe that it would be more reasonable to do it in a separate Discussion Section. Therefore we preferred to do so.
[7]. XRD results are included in Section 3.1., so it not good to claim the existence of carbide phase before discussing or presenting the results of XRD.
A: Thank you. The XRD results on as-clad coatings were extracted and moved to the Section3.1 по
[8]. Why the legend in Figures 4 is tilting, is it a scanned image or plotted?
A: Corrected
[9]. Is it the right heading for Section 3.4? These are microstructures or macro surface images?
A: Thank you. Revised to read:” 3.4. Morphology of the worn surfaces”
[10].The values written in Fig. 6(a’, b’, c’, d’) are blur, and how you plotted these roughness plot?
A: The images have been replaced for those with higher sharpness. The Fig. 6(a’, b’, c’, d’) images were obtained using the Olympus OLS LEXT 4100 laser scanning microscope software
[11].Conclusion section is completely vague, which needs to be rearranged and rewritten.
A: Rewritten

Round 2
Reviewer 1 Report
The authors have strongly improved their original manuzcript, which now certainly
Can be recommended for publication.